# The Effect of Magnetically Induced Local Structure and Volume Fraction on the Electromagnetic Properties of Elastomer Samples with Ferrofluid Droplet Inserts

**Catalin N. Marin [1] and Iosif Malaescu [1,2,\*]**

1   Department of Physics, West University of Timisoara, Bd. V. Parvan No. 4, 300223 Timisoara, Romania; catalin.marin@e-uvt.ro
2   Institute for Advanced Environmental Research, West University of Timisoara (ICAM-WUT), Oituz Str., No. 4, 300086 Timisoara, Romania
\*   Correspondence: iosif.malaescu@e-uvt.ro

**Abstract:** The magnetic permeability ($\mu$), dielectric permittivity ($\varepsilon$) and electrical conductivity ($\sigma$) of six elastomer samples obtained by mixing silicone rubber (RTV-530) with a kerosene-based ferrofluid in different volume fractions ($\varphi$), 1.31%, 2.59% and 3.84%, were determined using complex impedance measurements over a frequency range of *500 Hz–2 MHz*. Three samples ($A_0$, $B_0$ and $C_0$) were manufactured in the absence of a magnetic field, and the other three samples ($A_h$, $B_h$ and $C_h$) were manufactured in the presence of a magnetic field, $H = 43$ kA/m. The component $\mu''$ of the complex effective magnetic permeability of all samples presents a maximum at a frequency, $f_{max}$, that moves to higher values by increasing $\varphi$, with this maximum being attributed to Brownian relaxation processes. The conductivity spectrum, $\sigma(f)$, of all samples follows the Jonscher universal law, which allows for both the determination of the static conductivity, $\sigma_{DC}$, and the barrier energy of the electrical conduction process, $W_m$. For the same $\varphi$, $W_m$ is lower, and $\sigma_{DC}$ is higher in the samples $A_h$, $B_h$ and $C_h$ than in the samples $A_0$, $B_0$ and $C_0$. The performed study is useful in manufacturing elastomers with predetermined properties and for possible applications such as magneto-dielectric flexible electronic devices, which can be controlled by the volume fraction of particles or by an external magnetic field.

**Keywords:** elastomer; ferrofluid; complex magnetic permeability; complex dielectric permittivity; electrical conductivity

## 1. Introduction

A composite material is a system of two or more components with different properties that are mixed to obtain a material with increased performance compared with its components. In recent years, there has been increasing interest in research on composite materials with elastomers so they can be used in various fields of science and technology [1–5].

A special interest has recently been observed in the development of multifunctional polymeric nanocomposites consisting of metal particles/oxides dispersed in a polymeric matrix [6]. The electrical properties of these composites depend on both the nature, size and concentration of the dispersed particles and the properties of the polymer matrix [7].

In [8], the authors show that, via the mixing of natural rubber (NR) with metal particles (Ni or Fe), elastomers with improved electrical and magnetic properties can be obtained. Epoxy composites filled with graphite nanoplatelets and magnetite [9] or with carbon nanotubes [10] represent new composite materials with important microwave properties. Also, epoxy resin powder, when mixed with ferrite powders, leads to the obtainment of composite magnets of the ferrite/polymer type, with magnetic properties different from conventional magnets, which can be used in multiple applications [11]. In recent years, another class of composites obtained with a combination of silicone rubber and iron oxide compounds has been much studied [12–14], and these composite materials, showing

excellent magneto-dielectric properties, are useful in technological applications [15,16]. Also, one intensively studied fluidic composite is magnetic fluid, or ferrofluid, which is defined as a biphasic system of single-domain nanoparticles (Fe, Co, $Fe_3O_4$, $Fe_2O_3$, $CoFe_2O_4$, etc.) dispersed in a basic liquid and stabilized with a surfactant to prevent sedimentation [17]. On the other hand, ferrofluid is considered a composite with magneto-dielectric properties that is influenced by the presence of magnetic fields [18,19].

In this paper, we report the manufacturing of six composite samples obtained by mixing silicone rubber (RTV-530) with a kerosene-based ferrofluid sample with magnetite particles in three volume fractions ($\varphi$) (1.31%, 2.59% and 3.84%). For each volume fraction, $\varphi$, two samples were obtained: one polymerized in the presence of a static magnetic field, $H = 43$ kA/m, and another polymerized in the absence of the magnetic field. Based on dynamic measurements over a range of 500 Hz–2 MHz, the effect of volume fraction and polymerization in a magnetic field on both the complex effective magnetic permeability ($\mu = \mu' - i\mu''$) and complex dielectric permittivity ($\varepsilon = \varepsilon' - i\varepsilon''$) of the elastomer samples was investigated. The measurements of complex dielectric permittivity allowed for the determination of the electrical conductivity ($\sigma$) of the manufactured samples. The obtained results were discussed considering the theoretical model CBH (*correlated barrier hopping*), which is useful in possible technological applications.

## 2. Obtaining and Characterizing Samples

Six elastomer-type composite samples consisting of commercial RTV-530 silicone rubber (SR) from Prochima [20] with a density of $\rho_{SR} = 1.3$ g/cm$^3$ and a kerosene-based ferrofluid (EFH 1 type from Ferrotech) [21] with magnetite particles stabilized with oleic acid and a density of $\rho_{FM} = 1.21$ g/cm$^3$ were manufactured by mixing them. Silicone rubber has two components (A and B) and is a non-toxic elastomer with medium hardness and elasticity that can be used even as bolus material in radiotherapy [22].

In order to manufacture the composite samples, we mixed the same quantity, $M_{SR}$, of silicone rubber (equal quantities, $M_{SR}$/2, of each component, A and B) with a small amount, MFM, of ferrofluid (a few drops), thus obtaining three samples, which differ in the volume fraction, $\varphi$, of the ferrofluid in the composite. The mixture thus formed was placed in a parallelepiped mold and pressed continuously for 2–3 min by hand (just like a dough) until it took the shape of the mold, and after 24 h, the composite samples were obtained and took the shape of a square parallelepiped plate with a side measuring *5* cm and a thickness of 0.1 cm. This polymerization/hardening of the sample mixture was performed in the absence of a magnetic field, and the composite samples obtained are denoted by sample $A_0$, sample $B_0$ and sample $C_0$. Also, three other composite samples with the same volume fractions, $\varphi$, were polymerized in the presence of a magnetic field, $H = 43$ kA/m, and are denoted as sample $A_h$, sample $B_h$ and sample $C_h$. For this, the parallelepiped mold, which contains the mixture corresponding to samples $A_h$, $B_h$ and $C_h$, was fixed between the magnetic poles, N and S, of a Weiss-type electromagnet powered by a direct current source. The orientation of the magnetic field, $H$, was parallel to the surface of the sample, and the value of the magnetic field, $H$, was measured with the aid of a Hall probe from a Gauss meter.

The following quantities of materials were used to obtain the composite samples:

- For samples $A_0$ and $A_h$: $M_{FM} = 0.05$ g and $M_{SR} = 4$ g (2 g of each component of the silicon rubber, A and B);
- For samples $B_0$ and $B_h$: $M_{FM} = 0.10$ g and $M_{SR} = 4$ g (2 g of each component of the silicon rubber, A and B);
- For samples $C_0$ and $C_h$: $M_{FM} = 0.15$ g and $M_{SR} = 4$ g (2 g of each component of the silicon rubber, A and B).

Knowing the density and mass of the components of the elastomeric composite samples (ferrofluid and silicone rubber) shows us the total volume of the sample, $V_{tot} = V_{SR} + V_{FM}$, and then the volume fraction, $\varphi = V_{FM}/V_{tot}$, of the ferrofluid in the composite samples. We obtained the following values: $\varphi_1 = 1.31\%$, $\varphi_2 = 2.59\%$ and $\varphi_3 = 3.84\%$.

Figure 1 shows optical microscopy images of the composite samples, obtained both in the absence of the magnetic field (samples $A_0$, $B_0$ and $C_0$ in Figure 1a–c) and in the presence of a magnetic field, $H$ (samples $A_h$, $B_h$ and $C_h$ in Figure 1d–f). As can be seen in Figure 1, when the sample preparation takes place in the presence of a magnetic field (Figure 1d–f), the droplets of the ferrofluid stretch along the direction of the magnetic field lines, as well as the magnetite nanoparticles aggregate, thus forming field-induced structures, while in the absence of the magnetic field (Figure 1a–c), the droplets are approximately spherical in shape, oriented randomly in the entire volume of the composite material.

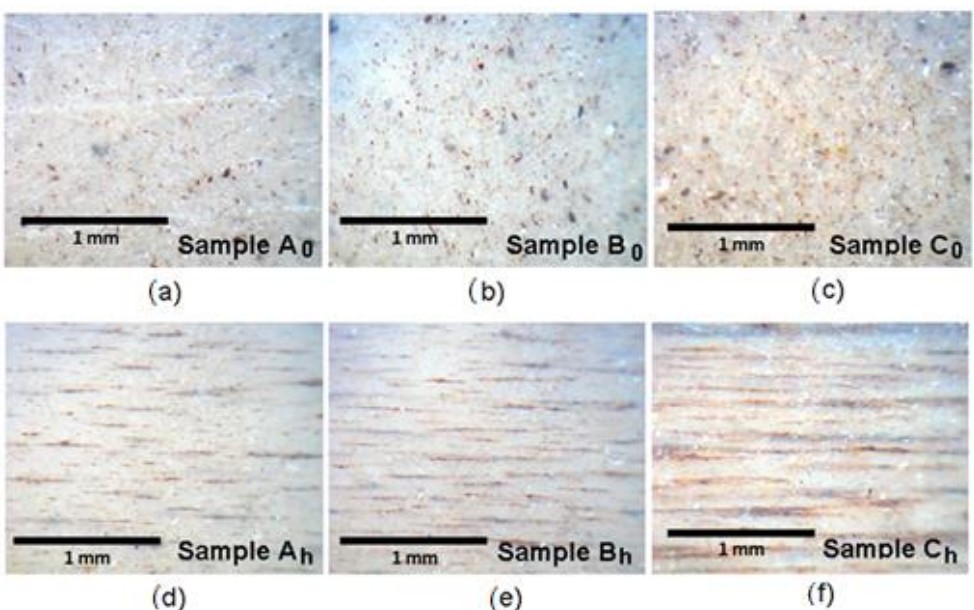

**Figure 1.** Images of the composite samples consisting of silicone rubber with ferrofluid: samples $A_0$ (**a**), $B_0$ (**b**) and $C_0$ (**c**) obtained in the absence of magnetic field; samples $A_h$ (**d**), $B_h$ (**e**) and $C_h$ (**f**) obtained in the presence of a magnetic field, $H$ = 43 kA/m.

The dependence of the magnetization, $M$, on the intensity of the magnetic field, $H$, for the ferrofluid used in the elastomeric composite samples is shown in Figure 2.

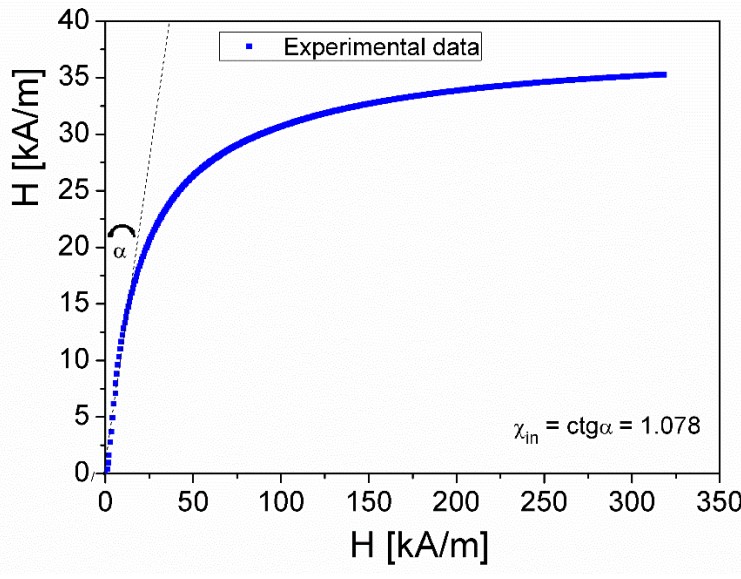

**Figure 2.** The magnetization curve, *M (H)*, of ferrofluid.

This dependence was obtained by using the inductive method with a hysteresis-graph [23] at low frequency (50 Hz). The dependence, *M(H)*, from Figure 2 obeys a Langevin-type law, which indicates the superparamagnetic behavior of the ferrofluid [17]. From Figure 2 and using Chantrell's magneto-granulometric analysis method [24], we determined the values of the following magnetic parameters of the ferrofluid: the saturation magnetization, $M\infty = 39.5$ kA/m; the initial susceptibility, $\chi_{in} = 1.078$; the concentration of particles, $n = 10.49 \cdot 10^{22}$ m$^{-3}$; and the mean magnetic diameter of the particles, $d_m = 11.74$ nm.

## 3. Results and Discussion

### 3.1. Investigation of Complex Effective Magnetic Permeability

The frequency dependencies of the real ($\mu'$) and imaginary ($\mu''$) components of the complex effective magnetic permeability of each sample were measured with the complex impedance method. For this, both the resistance, *R* (or $R_0$), and the inductive reactance, *X* (or $X_0$), of a solenoid with a sample as a magnetic core (or empty) connected to an RLC meter (Agilent type E-4980A) were measured at frequencies between 0.5 kHz and 2 MHz. $\mu'$ and $\mu''$ were determined with Equation (1) [25].

$$\mu' = \frac{X}{X_0} \qquad\qquad \mu'' = \frac{R - R_0}{X} \tag{1}$$

The elastomeric composite samples were obtained in the form of a rectangular plate (see Figure 1) and were rolled up like a scroll, thus obtaining a cylindrical shape for the sample, which was then inserted into the suitable measuring coil to fill the entire inner space of the coil as well as possible. In this way, what was measured represented the effective permeability of the sample. In the case of samples $A_h$, $B_h$ and $C_h$, obtained in the presence of an external magnetic field (see Figure 1d–f), the rolling up of the rectangular samples was made parallel to the field-induced structures, thus obtaining a cylindrical shape for the sample, in which the microstructures were arranged parallel to the axis of the obtained cylinder. In this way, the magnetic probing field of the coil is oriented parallel to the field-induced tubular structures obtained in samples $A_h$, $B_h$ and $C_h$ via their polymerization in a static magnetic field. The obtained results for the composite samples are presented in Figure 3.

In Figure 3a,b, it can be observed that, at a constant value of the volume fraction, $\varphi$, the real component, $\mu'$, of the complex effective magnetic permeability remains approximately constant with the frequency change. Also, for all samples, $\mu'$ increases by increasing the volume fraction, $\varphi$, of the magnetite particles dispersed in the composite. It should be noted that the values of $\mu'$ corresponding to samples $A_h$, $B_h$ and $C_h$ (Figure 3b), obtained in the presence of a magnetic field, *H*, are higher than those corresponding to samples $A_0$, $B_0$ and $C_0$ (Figure 3a), obtained in the absence of the magnetic field, *H*, at all values of the volume fraction, $\varphi$. This result shows that preparing such samples by mixing a ferrofluid with silicone rubber in the presence of an external magnetic field, *H*, leads to the obtainment of composite samples with improved magnetic properties that can be controlled by a magnetic field, *H*, and a volume fraction, $\varphi$.

The imaginary component, $\mu''$, of complex effective magnetic permeability has a local maximum (Figure 3) at a frequency, $f_{max}$, that depends on the volume fraction, $\varphi$, for each composite sample. The existence of this local maximum indicates a relaxation process in the composite elastomeric samples in the investigated frequency range, which is characterized by the relaxation time, $\tau$.

From Debye's theory [26], it is known that the relaxation time, $\tau$, is related to the frequency, $f_{max}$, at which $\mu''$ is at maximum via the following relation:

$$2\pi f_{max}\tau = 1 \tag{2}$$

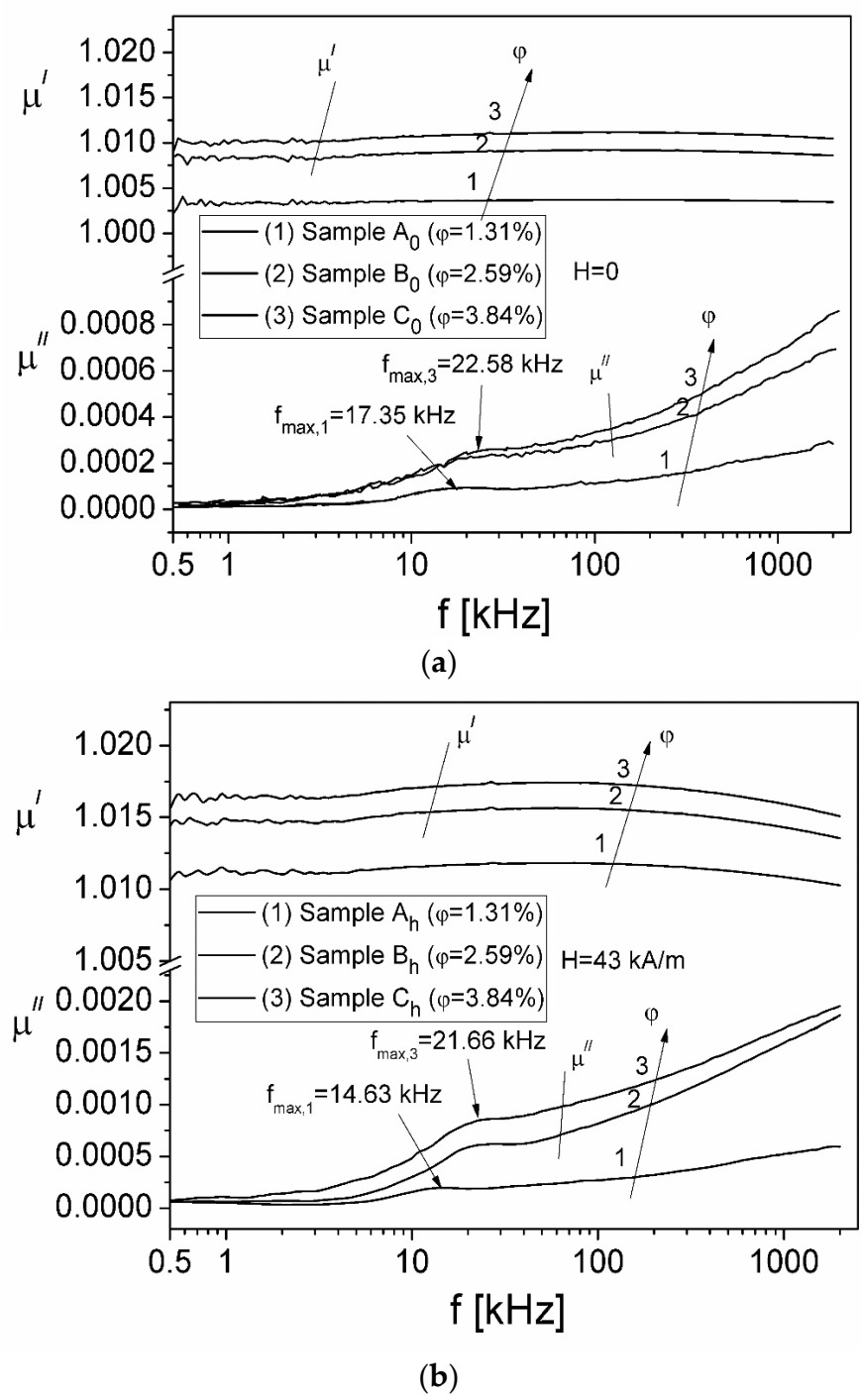

**Figure 3.** Variation in the frequency of the $\mu'$ and $\mu''$ components of complex effective magnetic permeability for the elastomeric composite samples: (**a**) $A_0$, $B_0$ and $C_0$; (**b**) $A_h$, $B_h$ and $C_h$.

Considering the experimental values, $f_{max}$, from Figure 3a,b and using Equation (2), the corresponding values of the relaxation times were computed, resulting in the following values: $\tau_{(A0)} = 9.18$ µs, $\tau_{(B0)} = 7.46$ µs and $\tau_{(C0)} = 7.05$ µs for samples $A_0$, $B_0$ and $C_0$ and $\tau_{(Ah)} = 10.88$ µs, $\tau_{(Bh)} = 7.96$ µs and $\tau_{(Ch)} = 7.35$ µs for samples $A_h$, $B_h$ and $C_h$, respectively. The dependence on the volume fraction, $\varphi$, of the obtained relaxation times, $\tau$, is shown in Figure 4.

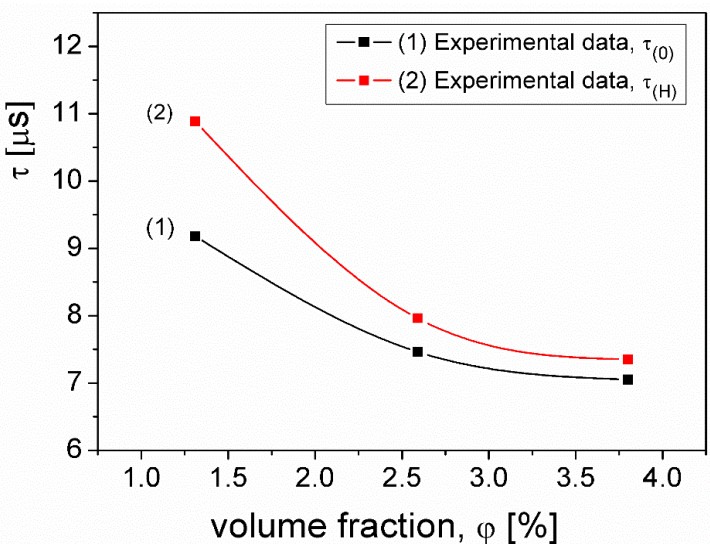

**Figure 4.** Volume fraction dependence of the relaxation times, $\tau(\varphi)$, for composite samples: $\tau_{(0)}$ for samples $A_0$, $B_0$ and $C_0$ and $\tau_{(H)}$ for the samples $A_h$, $B_h$ and $C_h$ (the line is a spline interpolation of the experimental points).

The obtained maxima of $\mu''$ in Figure 3a,b could be attributed to either the Néel relaxation process or the Brownian relaxation process. In the case of the Néel relaxation process, the magnetic movements of the particles rotate inside the particles, and the particles remain fixed in the composite [27]; the relaxation time, $\tau_N$, is provided by the relation

$$\tau_N = \tau_0 \exp\left(\frac{KV_m}{kT}\right) \tag{3}$$

where $\tau_0$ is a constant that can take values between $10^{-12}$ s and $10^{-9}$ s, depending on the material from which the particles are made. For magnetite, it is usually considered $\tau_0 = 10^{-9}$ s [27,28]. $T$ is the absolute temperature; $k$ is Boltzmann's constant; $V_m$ is the magnetic volume of a particle; and $K$ is the anisotropy constant of particles.

The Brownian relaxation process is correlated to the particle's rotation, or the rotation of particle aggregates, in the carrier liquid [27], as characterized by Brownian relaxation time, $\tau_B$, which is provided by the following equation:

$$\tau_B = \frac{\pi \eta D_h^3}{2kT} \tag{4}$$

Here, $D_h$ is the hydrodynamic diameter of the particle or the aggregate, and $\eta$ is the dynamic viscosity of the carrier liquid.

Taking into account the values of the relaxation times obtained for the investigated composite elastomeric samples and the $d_m$ value of the mean magnetic diameter of the particles, if we consider that the relaxation process would be a Néel type, the anisotropy constant, $K$, of the magnetic particles can be computed with Relation (3). The following values were obtained: $K_{(A0)} = 4.46 \cdot 10^4$ J/m$^3$, $K_{(B0)} = 4.36 \cdot 10^4$ J/m$^3$ and $K_{(C0)} = 4.33 \cdot 10^4$ J/m$^3$ for samples $A_0$, $B_0$ and $C_0$ and $K_{(Ah)} = 4.54 \cdot 10^4$ J/m$^3$, $K_{(Bh)} = 4.38 \cdot 10^4$ J/m$^3$ and $K_{(Ch)} = 4.35 \cdot 10^4$ J/m$^3$ for samples $A_h$, $B_h$ and $C_h$. The results thus obtained for the anisotropy constant, $K$, of the magnetite particles in the composite elastomeric samples far exceed the $K$ values corresponding to magnetite particles $(1.1 \cdot 10^4 - 1.5 \cdot 10^4)$ J/m$^3$ [29,30]. This allows us to draw the conclusion that the relaxation process afferent to the local maxima of $\mu''$ (Figure 3) cannot be considered a Néel relaxation process.

If we assume that the relaxation process is Brownian, by replacing the values of the relaxation time corresponding to all the investigated composite samples in Equation (4) and considering the value $\eta = 1.2 \cdot 10^{-3}$ Pa·s for the viscosity of the carrier liquid (kerosene)

and the constant room temperature, T = 300 K, at which the measurements were made, we can determine the hydrodynamic diameter, $D_h$, of the particles in the samples. The values obtained are $D_{h,A0}$ = 27.13 nm, $D_{h,B0}$ = 25.32 nm and $D_{h,C0}$ = 24.84 nm for samples $A_0$, $B_0$ and $C_0$ and $D_{h,Ah}$ = 28.71 nm, $D_{h,Bh}$ = 25.87 nm and $D_{h,Ch}$ = 25.20 nm for samples $A_h$, $B_h$ and $C_h$, respectively. The values determined for the hydrodynamic diameter, $D_h$, show that, in all samples, aggregates of 2–3 particles rotate as a single structure in the carrier liquid of the ferrofluid within the droplet inserts from the composite. So, the maximum of the imaginary component, $\mu''$, from Figure 3a,b is due to the Brownian relaxation process in the composite, and the ferrofluid droplet inserts are still present in the composite after polymerization.

To support the statement that the relaxation maximum of the elastomeric composite samples is the Brownian relaxation maximum (given the rotation of aggregates in the carrier liquid of the ferrofluid), we performed complex magnetic permeability measurements for the ferrofluid as well. The obtained result is presented in Figure 5. In Figure 5, it can be observed that at a frequency of 13.76 kHz, $\mu''$ has a shoulder similar to the elastomeric composite samples. Moreover, when applying a low-intensity field, H = 4 kA/m, can be observed that the shoulder turns into a well-defined relaxation maximum at a frequency of 11.39 kHz. The behavior of the ferrofluid confirms the fact that the $\mu''$ maximum in the composite elastomer samples is a Brownian relaxation maximum in the ferrofluid.

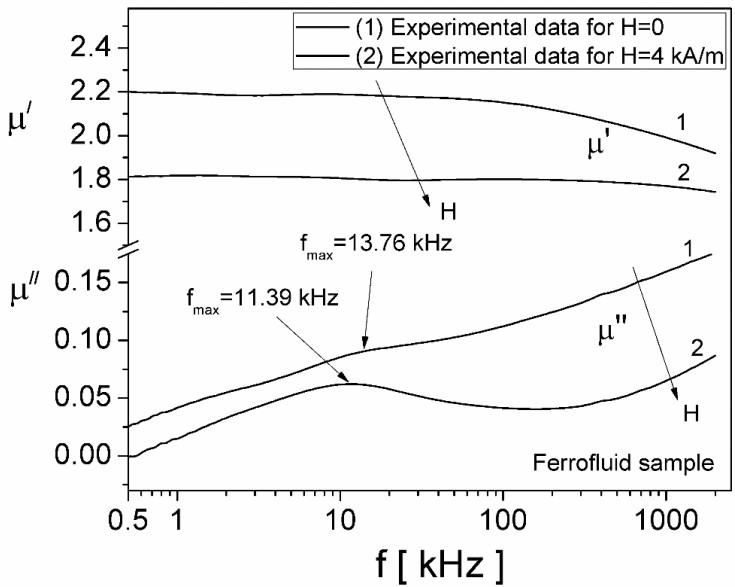

**Figure 5.** The frequency dependence of the real, $\mu'$, and imaginary, $\mu''$, components of the complex magnetic permeability of the ferrofluid sample, both in no magnetic field (*H = 0*) and in the presence of a field (*H ≠ 0*).

Also, Figure 1 shows that ferrofluid droplet inserts are present in all samples. For samples $A_0$, $B_0$ and $C_0$, the droplets are approximately spherical in shape, and for the samples polymerized in the magnetic field, the droplets are elongated along the magnetic field lines.

### 3.2. Investigation of Complex Dielectric Permittivity

The real component, $\varepsilon'$, and imaginary component, $\varepsilon''$, of the complex dielectric permittivity were determined over a frequency range of 500 Hz–2 MHz. For this, each composite sample was placed in a planar capacitor with circular plates with a diameter of 4 cm and a distance between plates of *d* = 1 mm. The capacitor with a composite sample was connected to an RLC meter, and the electric field between the plates of the capacitor was perpendicular to the sample. In the case of samples Ah, Bh and Ch, obtained in the

presence of an external magnetic field (see Figure 1d–f), the electric field between the plates of the capacitor was perpendicular to the field-induced tubular structures. For a fixed frequency, $f$, the RLC meter indicated resistance, $R$, and reactance, $X$, in the presence of a composite sample within the capacitor and resistance, $R_0$, and reactance, $X_0$, in the absence of a sample in the capacitor. Components $\varepsilon'$ and $\varepsilon''$ of the complex dielectric permittivity were determined with the following relations [31,32]:

$$\varepsilon' = \frac{X_0}{X} \qquad\qquad \varepsilon'' = X_0\left(\frac{1}{R} - \frac{1}{R_0}\right) \tag{5}$$

Figure 6 shows the frequency dependence of the real ($\varepsilon'$) and imaginary ($\varepsilon''$) components in the frequency range 500 Hz–2 MHz at different values of volume fraction, $\varphi$, for the magnetite particles.

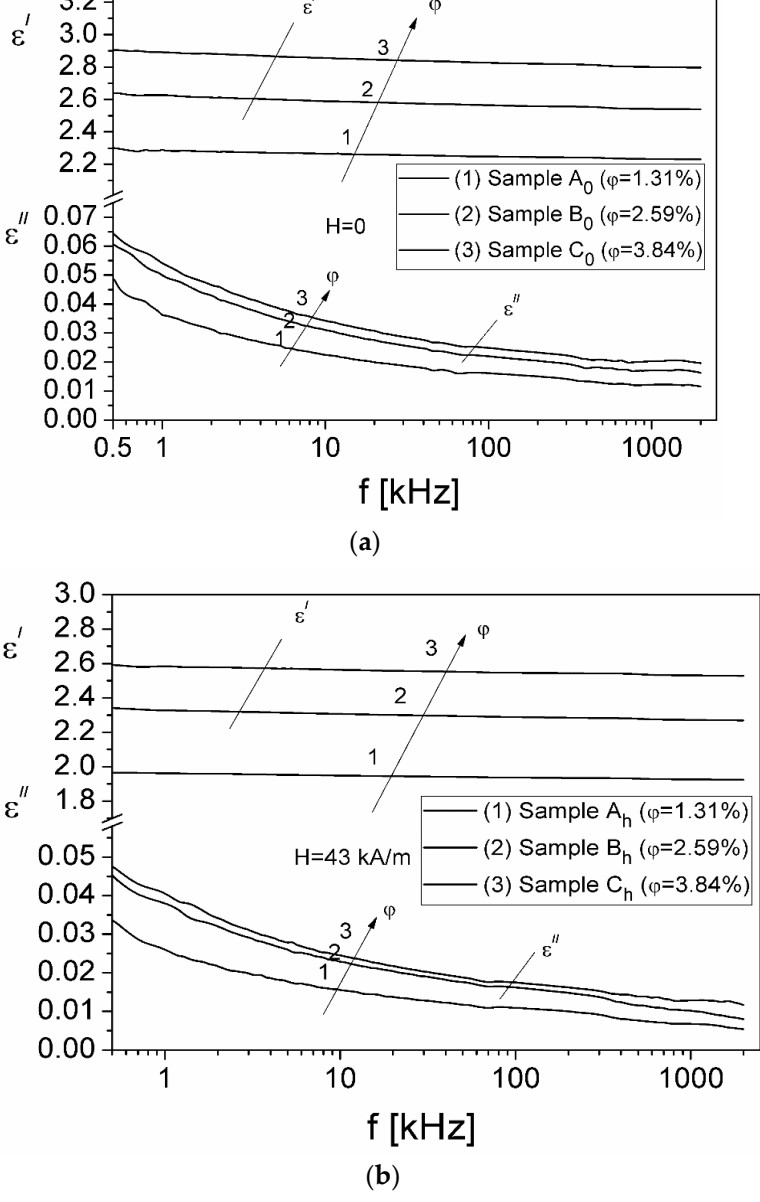

**Figure 6.** The frequency dependence of the real, $\varepsilon'$, and imaginary, $\varepsilon''$, components of the complex dielectric permittivity of (**a**) composite samples $A_0$, $B_0$ and $C_0$ and (**b**) composite samples $A_h$, $B_h$ and $C_h$.

As can be observed in Figure 6a,b, at a constant value of the volume fraction, $\varphi$, the real component, $\varepsilon'$, of the complex dielectric permittivity remains approximately constant with the frequency change. Also, one can observe that $\varepsilon'$ increases from 2.3 to 2.9 (for samples $A_0$, $B_0$ and $C_0$—see Figure 6a) and from 2.0 to 2.6 (for samples $A_h$, $B_h$ and $C_h$—see Figure 6b) by increasing the volume fraction, $\varphi$, from 1.31% to 3.84%. The values of $\varepsilon'$ corresponding to samples $A_h$, $B_h$ and $C_h$ (Figure 6b), obtained in the presence of a magnetic field $H$, are lower than those corresponding to samples $A_0$, $B_0$ and $C_0$ (Figure 6a), obtained in the absence of a magnetic field, $H$, at all values of the volume fraction, $\varphi$. This result can be correlated with a decrease in the equivalent electric capacity of the sample holder with samples $A_h$, $B_h$ and $C_h$ (polymerized in a magnetic field) versus that of the sample holder with samples $A_0$, $B_0$ and $C_0$ (polymerized in no magnetic field). A similar result for a ferrofluid sample was reported in Ref. [31]. When inserting $A_h$, $B_h$ and $C_h$ samples between capacitor armatures, the electric field lines will be perpendicular to the microstructures induced by polymerization in a magnetic field. As a result, such a structure leads to a decrease in the equivalent capacity and, therefore, the dielectric permittivity, in accordance with the Wigner limits of the permittivity of composite materials [33].

For a constant value of the volume fraction, $\varphi$, the imaginary component, $\varepsilon''$, of the complex dielectric permittivity decreases with increasing frequency, $f$, both for samples $A_0$, $B_0$ and $C_0$ and for samples $A_h$, $B_h$ and $C_h$ (see Figure 6a,b). Also, at the same value of the volume fraction, $\varphi$, the values of $\varepsilon''$ corresponding to samples $A_h$, $B_h$ and $C_h$ (Figure 6b) are smaller than those corresponding to samples $A_0$, $B_0$ and $C_0$ at any given frequency.

This result shows that preparing such samples by mixing a ferrofluid with silicone rubber in the presence of an external magnetic field leads to the obtainment of composite samples with different dielectric properties that can be controlled by a magnetic field, $H$ and by a volume fraction, $\varphi$.

*3.3. DC and AC Conductivity*

It is known that, for the study of composite materials, an important parameter is electrical conductivity, $\sigma$, which can be determined from the dielectric data of permittivity [33,34] with the following relation:

$$\sigma = 2\pi f \varepsilon_0 \varepsilon'' \tag{6}$$

Taking into account the experimental values of $\varepsilon''$ obtained for the composite elastomeric samples (Figure 6), with Equation (6), we computed the conductivity, $\sigma$, whose frequency dependence is shown in Figure 7 for all investigated samples. Knowing the conductivity, $\sigma$ is very useful in understanding the transport of electric charges in the studied material [34] and for its applications.

In Figure 7, it can be observed that the conductivity spectrum, $\sigma(f)$, presents two regions: (1) a region in which $\sigma$ remains constant with the frequency, corresponding to DC-conductivity ($\sigma_{dc}$), and (2) a dispersion region, where $\sigma$ depends on frequency, corresponding to AC-conductivity ($\sigma_{ac}$). In other papers [9,35], a similar conductivity frequency dependence was obtained for other composite samples using a combination of $Fe_3O_4$ nanoparticles or graphite nanoplatelets and a polymer. The frequency behavior of the electrical conductivity of the elastomer composite samples (as seen in Figure 7) agrees with Jonscher's universal law [36]:

$$\sigma(\omega) = \sigma_{DC} + \sigma_{AC} \tag{7}$$

The values of static conductivity, $\sigma_{DC}$, remain approximately constant with frequency, up to about 30 kHz, for each volume fraction, $\varphi$, both for composite samples $A_0$, $B_0$ and $C_0$ (Figure 6a) and for samples $A_h$, $B_h$ and $C_h$ (Figure 6b); the obtained $\sigma_{DC}$ values are listed in Table 1.

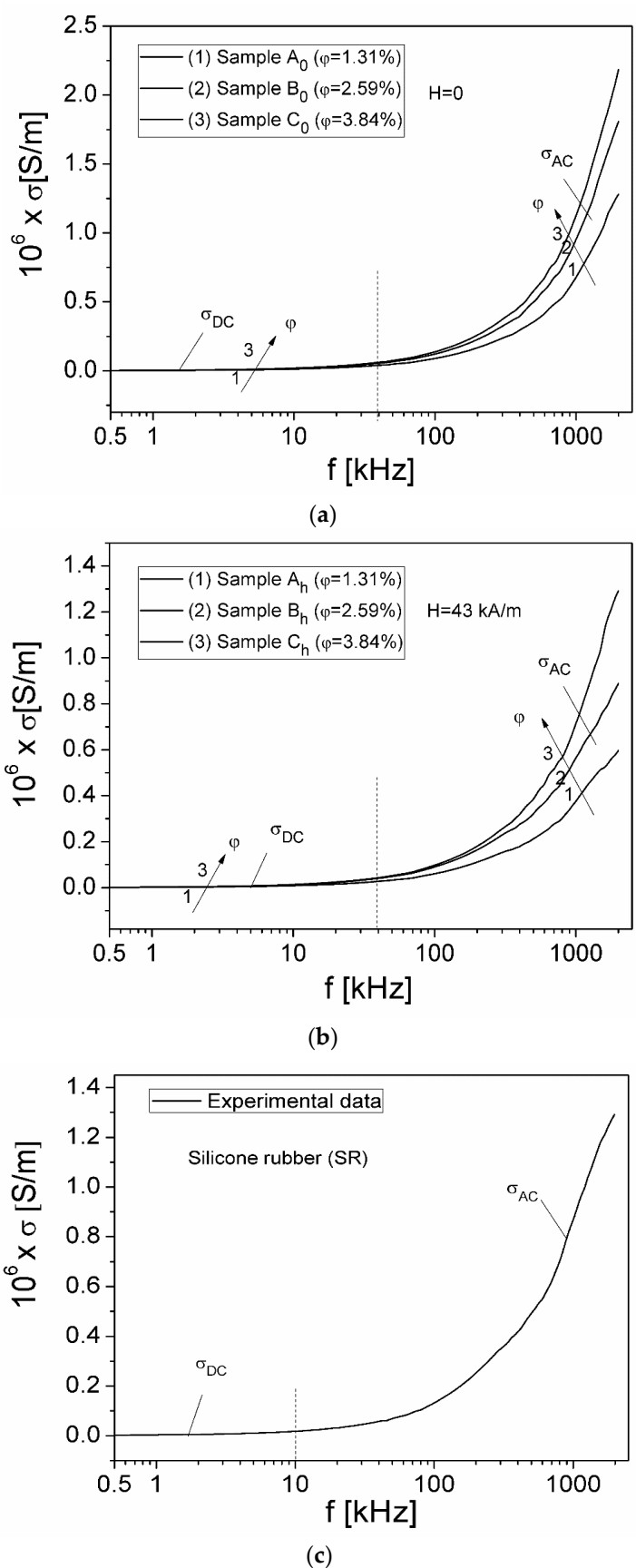

**Figure 7.** The frequency dependence of the conductivity, $\sigma$, of samples $A_0$, $B_0$ and $C_0$ (**a**); samples $A_h$, $B_h$ and $C_h$ (**b**); and silicone rubber (**c**).

**Table 1.** The parameters of composite samples determined from measurements.

| Samples | $A_0$ | $B_0$ | $C_0$ | $A_h$ | $B_h$ | $C_h$ |
|---|---|---|---|---|---|---|
| | $\varphi = 1.31\%$ | $\varphi = 2.59\%$ | $\varphi = 3.84\%$ | $\varphi = 1.31\%$ | $\varphi = 2.59\%$ | $\varphi = 3.84\%$ |
| **Parameters** | | $H = 0$ | | | $H = 43$ kA/m | |
| $\sigma_{DC}$ (S/m) | $4.26 \cdot 10^{-9}$ | $9.40 \cdot 10^{-9}$ | $1.03 \cdot 10^{-8}$ | $4.93 \cdot 10^{-9}$ | $1.73 \cdot 10^{-8}$ | $1.86 \cdot 10^{-8}$ |
| $n$ | 0.897 | 0.915 | 0.938 | 0.751 | 0.807 | 0.872 |
| $A$ (S/m) | $5.42 \cdot 10^{-13}$ | $5.72 \cdot 10^{-13}$ | $4.73 \cdot 10^{-13}$ | $28.7 \cdot 10^{-13}$ | $20.4 \cdot 10^{-13}$ | $8.56 \cdot 10^{-13}$ |
| $W_m$ (eV) | 1.51 | 1.83 | 2.51 | 0.62 | 0.81 | 1.22 |

Also, as seen in Figure 7c, we determined the static conductivity, $\sigma_{DC}$, of silicone rubber (SR), obtaining the value $\sigma_{DC} = 1.4 \times 10^{-9}$ S/m. As a result, by adding ferrofluid to the silicone rubber (SR), the static conductivity, $\sigma_{DC}$, of the elastomeric composite samples was increased compared with the $\sigma_{DC}$ value of the silicone rubber, which was all the higher in the volume fraction of the ferrofluid (see Table 1).

In Table 1, it can be observed that, by increasing the volume fraction, $\varphi$, of the particles, the $\sigma_{DC}$ conductivity increases for all composite samples. Also, the values of $\sigma_{DC}$ corresponding to samples manufactured in the presence of a magnetic field (samples $A_h$, $B_h$ and $C_h$) are higher than the $\sigma_{DC}$ values of samples $A_0$, $B_0$ and $C_0$, manufactured in the absence of a magnetic field. Therefore, the $\sigma_{DC}$ conductivity of the composite samples is correlated with the sample manufacturing process. When sample preparation takes place in the presence of a magnetic field, the magnetite particles from ferrofluid align in the direction of the magnetic field, forming parallel chains of particles, which leads to an increase in conductivity, $\sigma_{DC}$, with respect to the $\sigma_{DC}$ of samples prepared in the absence of a magnetic field when the particles are randomly oriented in the entire volume of the elastomer composite material (see Figure 1a–c).

The $\sigma_{AC}$ component of conductivity depends on frequency—correlated with dielectric relaxation processes due to localized electric charge carriers from the composite samples—and is provided by the following equation:

$$\sigma_{AC} = A\omega^n \tag{8}$$

Here, $n$ is an exponent that is dependent on both frequency and temperature ($0 < n < 1$), and $A$ is a pre-exponential factor [37].

The logarithmation of Equation (8) leads to a linear dependence between $ln\sigma_{AC}$ and $ln\omega$, which is shown in Figure 8a for samples $A_0$, $B_0$ and $C_0$ and in Figure 8b for samples $A_h$, $B_h$ and $C_h$. Fitting the experimental dependencies, $ln(\sigma_{AC})(ln(\omega))$, from Figure 8 with a straight line, we determined the exponent, $n$, and the parameter, $A$, corresponding to all the values of the volume fraction, $\varphi$. The values obtained are listed in Table 1. It can be observed that, for the same value of the volume fraction, $\varphi$, the values of the exponent, $n$, corresponding to samples $A_h$, $B_h$ and $C_h$ (obtained in the presence of a magnetic field $H$) are lower than the values, $n$, corresponding to samples $A_0$, $B_0$ and $C_0$ (obtained in the absence of a magnetic field).

To investigate the electrical conduction mechanism in the elastomeric composite samples, several theoretical models [38,39] can be applied, such as the commonly used correlated-barrier-hopping (CBH) theoretical model [39]. According to the CBH model, the exponent, $n$, can be written in a first approximation as [39]

$$n = 1 - \frac{6kT}{W_m} \tag{9}$$

In Equation (9), $W_m$ represents the barrier energy [39,40]. Using Relation (9), and the values of $n$, we determined the barrier energy of the electrical conduction process of each investigated sample. The obtained results for $W_m$ are shown in Table 1.

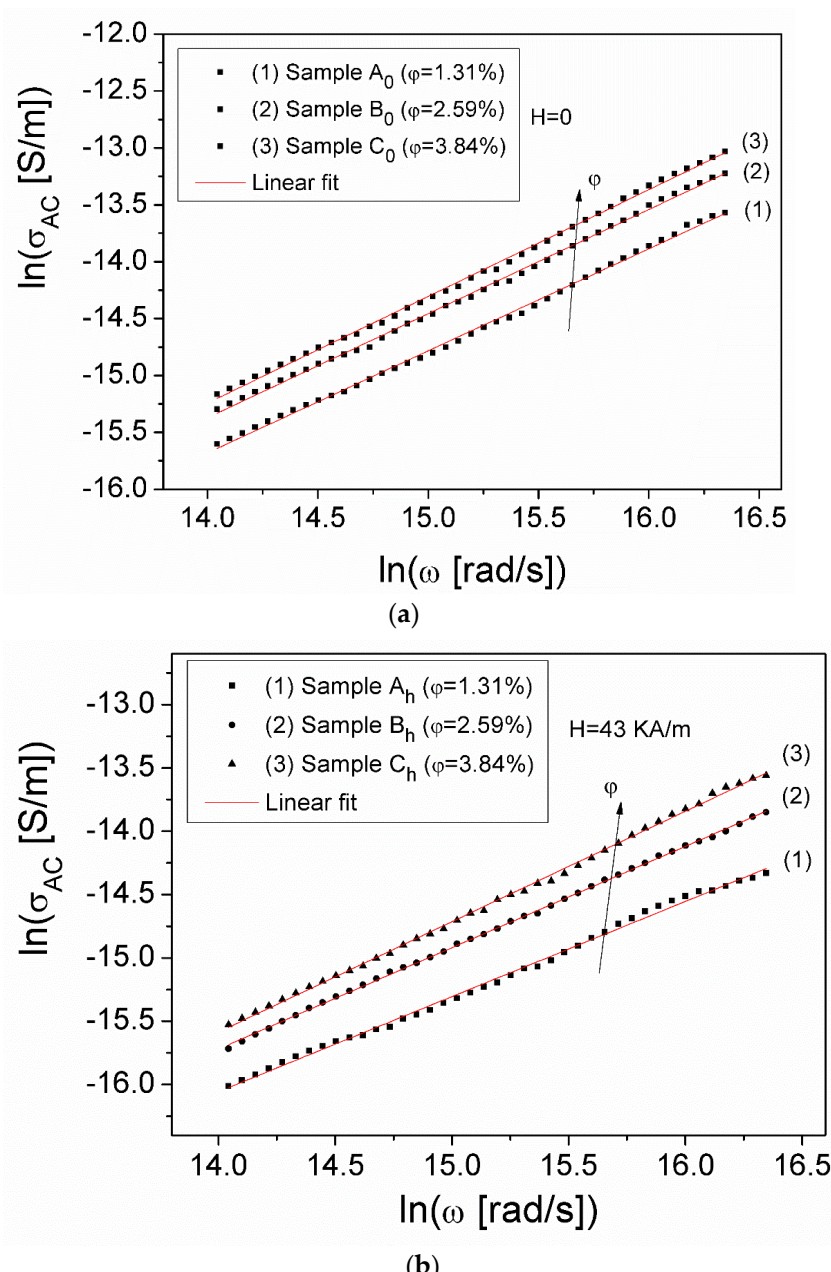

**Figure 8.** *ln$\sigma_{ac}$(ln$\omega$)* dependence for composite samples $A_0$, $B_0$ and $C_0$ (**a**) and samples $A_h$, $B_h$ and $C_h$ (**b**).

As can be seen from Table 1, an increase in the volume fraction, *φ*, of the particles leads to an increase in the barrier energy, $W_m$, of all composite samples. Also, the $W_m$ values corresponding to the samples manufactured in the presence of a magnetic field (samples $A_h$, $B_h$ and $C_h$) are lower than the $W_m$ values of samples $A_0$, $B_0$ and $C_0$, manufactured in the absence of a magnetic field. Therefore, a decrease in the barrier energy, $W_m$, of samples $A_h$, $B_h$ and $C_h$ compared with the barrier energy of samples $A_0$, $B_0$ and $C_0$ will lead to an increase in the number of charge carriers that will be able to participate in the electrical conduction of these samples, which determines an increase in their conductivity, as we achieved experimentally (see Table 1).

## 4. Conclusions

In this paper, six composite samples were manufactured both in the absence and in the presence of a static magnetic field by mixing silicone rubber (RTV-530) with a kerosene-

based ferrofluid with magnetite particles in different volume fractions of particles, $\varphi$ (1.31%, 2.59% and 3.84%). Based on the complex impedance measurements, in a low-frequency field over a range of 0.5 kHz–2 MHz, the complex effective magnetic permeability, $\mu = \mu' - i\mu''$; the complex dielectric permittivity, $\varepsilon = \varepsilon - i\varepsilon''$; and the electrical conductivity, $\sigma$, of all the composite samples were determined. The obtained results for the complex effective magnetic permeability of the elastomeric composite samples show that the imaginary component, $\mu''$, has a maximum at low frequencies between 0.1 and 0.3 kHz, both for samples manufactured in the absence of a magnetic field (samples $A_0$, $B_0$ and $C_0$) and for samples manufactured in the presence of a magnetic field, $H$ (samples $A_h$, $B_h$ and $C_h$). This maximum is attributed to the Brownian relaxation process within the ferrofluid droplet inserts from the composites. Using the experimental results for the complex dielectric permittivity, the conductivity spectra, $\sigma(f)$, of all the investigated samples were determined. The spectra, $\sigma(f)$, obey Jonscher's universal law, as they have two regions: a region in which $\sigma$ does not vary with frequency, corresponding to DC-conductivity ($\sigma_{DC}$), and a dispersion region where $\sigma$ rapidly increases with frequency, corresponding to AC-conductivity ($\sigma_{AC}$). An increase in the volume fraction of particles in the elastomeric composite samples, from $\varphi = 1.31\%$ to $x = 3.84\%$, leads to an increase in $\sigma_{DC}$ from $4.26 \cdot 10^{-9}$ S/m to $1.03 \cdot 10^{-8}$ S/m for samples $A_0$, $B_0$ and $C_0$ and from $4.93 \cdot 10^{-9}$ S/m to $1.86 \cdot 10^{-8}$ S/m for samples $A_h$, $B_h$ and $C_h$. Based on Jonscher's universal response and the CBH (correlated-barrier-hopping) theoretical model, we evaluated, for all composite samples, the energy barrier of the electrical conduction process, $W_m$. The results show that the $W_m$ values corresponding to the samples manufactured in the presence of a magnetic field (samples $A_h$, $B_h$ and $C_h$) are lower than the $W_m$ values of samples $A_0$, $B_0$ and $C_0$, manufactured in the absence of the magnetic field, for all values of the volume fraction, $\varphi$ (this result agrees with the increase in their conductivity, $\sigma_{DC}$). The results obtained are very useful for the manufacture of elastic composites with predetermined properties that can be tuned by changing the volume fraction of particles inside the composite or by modifying the local structure in the presence of an external magnetic field.

**Author Contributions:** Conceptualization, I.M. and C.N.M.; methodology, I.M.; software, C.N.M.; validation, I.M. and C.N.M.; formal analysis, C.N.M.; resources, C.N.M. and I.M.; investigation, C.N.M.; writing—original draft preparation, I.M.; writing—review and editing, C.N.M. and I.M.; visualization, I.M. and C.N.M.; supervision, C.N.M. and I.M. All authors have read and agreed to the published version of the manuscript.

**Funding:** This research received no external funding.

**Institutional Review Board Statement:** Not applicable.

**Informed Consent Statement:** Not applicable.

**Data Availability Statement:** Data will be made available upon request.

**Conflicts of Interest:** The authors declare no conflicts of interest.

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
