# Peer review of "The Effect of Magnetically Induced Local Structure and Volume Fraction on the Electromagnetic Properties of Elastomer Samples with Ferrofluid Droplet Inserts"

_magnetochemistry, doi:10.3390/magnetochemistry10010004_

Round 1
Reviewer 1 Report
Comments and Suggestions for Authors
The topic of the submitted manuscript on electromagnetic properties of elastomer with ferrofluid droplet is interesting and valuable to investigate. The authors expressed the idea and motivation of the study clearly in the introduction. In my opinion, the manuscript is suitable for publication in Magnetochemistry but the following comments and suggestions should be considered by the authors firstly:
1. In chapter 2, there is no information on the magnetic field source. Did author use permanent magnets or an electromagnet? I suggest to provide concise information on the magnetic setup configuration in order to increase the reproducibility of the described experiment.
2. The employed LCR meter allows measurements from 20 Hz. Did authors perform their measurements at frequencies below 500 Hz? Why did you restrict the lower frequency limit to 500 Hz?
3. The maximum in the imaginary part of the permeability was ascribed to the Brownian relaxation of the ferrofluid nanoparticles. To speak about the rotational relaxation of particles in the polymerized matrix is quite speculative. The authors explain that there are droplets of ferrofluids in the composite, but in the manuscript, there are practically no experimental proofs of that. From Fig. 1 one cannot understand the presence of any droplets. Anyway, if the maximum really reflects the Brownian relaxation of the ferrofluid droplets, then, I think that the same relaxation maximum should be observed when measuring the permeability directly of the ferrofluid. Can you confirm that?
4. From eq. (4) you obtained hydrodynamic diameter of the particles in the composite. What is the hydrodynamic diameter of the particles in the pure ferrofluid (not mixed with the SR)?
5. Could you provide any suggestion why the permittivity of Ah, Bh, Ch is smaller than that of A0, B0, C0?
6. What is the electrical conductivity of pure SR? It would be valuable to see the spectrum of electrical conductivity of the SR without the particles in order to understand the degree of the particle contribution to the conductivity of the composite.
Author Response
"Please see the attachment."

Reviewer 2 Report
Comments and Suggestions for Authors
This work is devoted to the experimental study of complex magnetic and dielectric permeability of rubber polymer samples with added ferrofluid. The results presented in the work are new and may be of scientific interest. However, there are a number of remarks on the work, the elimination of which is necessary before the possible publication of the paper.
1. From the description presented in the paper, it is not quite clear what the investigated composite material is. The title of the paper states that the material is an elastomer with drops of ferrofluid. However, further in the text, when describing the microstructure of the samples, it is stated that the sample represents magnetite nanoparticles distributed in the elastomer and their aggregates, which unite into chain structures in a magnetic field. When describing the mechanism of magnetization relaxation, drops are again mentioned. So in what form does the ferrofluid exist within the elastomer? Are they individual droplets or is there aggregation (violation of colloidal stability) of the ferrofluid when ferrofluid is added to the elastomer? To answer this question one can consider what happens to the structure of the sample after the magnetic field is removed while the polymer matrix is still liquid. If the ferrofluid forms droplets, they should return to a spherical shape after the magnetic field is removed.
2. The paper states that the sample was in the shape of a parallelepiped with a side of 5 cm and a thickness of 0.1 cm. How was the magnetic field directed relative to the parallelepiped during polymerization of the sample? Was the field homogeneous throughout the volume of the sample?
3. In describing the magnetic permeability measurement, nothing is said about what the measuring solenoid was. Taking into account that the sample had the shape of a rectangular plate, what was the geometry of the measuring solenoid and how was the sample under study placed in it? Did the sample fill the entire inner space of the measuring solenoid? If the sample did not fill the entire inner space of the measuring solenoid (which is most likely for a solid sample), then one should either determine a correction factor for the solenoid filling or speak of the effective (rather than true) permeability of the sample.
4. Also from the description it is not clear how the measuring field was directed when measuring the magnetic and dielectric permeability in relation to the direction of the magnetic field in which the sample was polymerized. This is very important because the anisotropic microgeometry of the sample resulting from the magnetic field leads to anisotropy of the macroscopic properties, and the magnetic and dielectric permeabilities measured in different directions will be different.
5. What explains the subsequent increase in the imaginary part of the magnetic permeability at measuring field frequencies above fmax? With Brownian relaxation described by Debye's theory, the imaginary part of the magnetic permeability should then decrease.
6. The paper does not give any physical interpretation of the effect of changing macroscopic properties of the samples as a result of the field action. It is clear that it is related to the formation of anisotropic microstructure of samples, but what is this relationship? I would like to see at least qualitative reasoning on this subject. This is all the more interesting because different macroscopic parameters of the samples change differently as a result of the field action. So magnetic permeability increases and dielectric permeability decreases as a result of magnetic field action, which is rather strange.
In conclusion, we note that the work requires substantial revision before it can be recommended for publication.
Comments on the Quality of English LanguageThe quality of the language is fine, just needs a little polishing in some places.
Author Response
"Please see the attachment."

Reviewer 3 Report
Comments and Suggestions for Authors
The authors present a systematic work where ferrofluid droplets are included in an elastomer with different volume fractions, and compare the magnetic, and dielectric properties of these composites which were prepared with and without the applying a magnetic field. The work is interesting, still it requires several points to be clarified.
1) In the sample preparation section several issues with the description, namely´
a. The authors state: “The mixture thus formed was placed in a parallelepiped mold and pressed continuously for 2-3 minutes until it takes the shape of mold”. The pressure used is not stated.
b. “and after24 hours the composite samples are obtained, having the shape of a square parallelepiped 75 plate with a side of 5 cm and a thickness of 0.1 cm.” what happens during the 24 hours? The sample is just at rest? Is the sample within the mold during the 24 hours? Is there any special care during the polymerization process?
c. Do the ferrofluid droplets interact with rubber? Do they remain in a liquid like state, or does the kerosene mix/diffuses to the silicon rubber and the iron oxide remains rigid in the rubber matrix?
d.
2) The authors state that figure 2 presents a static magnetization curve, still they use an AC (50 Hz) induction hysteresis-graph, which is not a true static magnetization (even though the frequency is not high).
3) In page 6, line 149 the authors see a “local maximum” and not a “maximum”, since the curve has a higher global maximum value.
4) Figure 4 shows a nonlinear curve connecting the points. Where does this curvature come from? Is it a eye guide or is there any fit law involved?
5) Line 176, the authors state that the attempt time tau0 is a constant that as a vale of 10^-9 s. As a matter of fact, this constant depends on the material and can have several values. Moreover, this constant, depending on the system, can have values between 10^-12 and 10^-9 s.
6) The previous point can also explain why the authors obtain different values of K from the ones of the literature. As a matter of fact, all this discussion involving the type of relaxation, as well as the estimation of K should be remade.
a. First, all the discussion is based on the magneto-granulometry analysis, which besides its intrinsic approximations, it is not clear how the authors did the analysis, nor the expressions used (despite referencing the paper by Chantrell)
b. Second, since it is not clear how the ferrofluid droplet interacted with the remaining elastometer, it is not clear that the viscosity considered is valid. Moreover, the viscosity is highly dependent on the temperature and no comment was made regarding this issue.
c. As mentioned before the attempt time used us too strict, hence it is not possible to rule out a Neel relaxation.
7) The introduction should also be a bit more extensive, namely including more elastometer containing iron oxides, such as :
Mohseni, Farzin, et al. "Bonded ferrite-based exchange-coupled nanocomposite magnet produced by Warm compaction." Journal of Physics D: Applied Physics 53.49 (2020): 494003.
Author Response
"Please see the attachment."

Round 2
Reviewer 1 Report
Comments and Suggestions for Authors
The authors have addressed all my comments and revised the manuscript accordingly.
Author Response
I would like to thank you for carefully reading the paper and for their comments and your suggestions so that all requested revisions was pertinent and important for improving the quality and understanding of our paper.
Reviewer 2 Report
Comments and Suggestions for Authors
The authors took into account the comments made earlier and modified the text of the article accordingly. In its current form, the article can be recommended for publication.
Comments on the Quality of English LanguageEnglish language fine.
Author Response

(The authors gave the same response as above.)

Reviewer 3 Report
Comments and Suggestions for Authors
After seeing the changes performed by the authors, and the reading their responses I consider that the quality of the manuscript has indeed been improved. Still, there are some points that the authors have yet to clarify.
Regarding point 1a, the authors answer with:
“The parallelepiped mold has the dimensions of the composite sample to be obtained (…) The silicone rubber used (RTV-530) (…) high pressure is not required to obtain the desired shapes/molds”
Despite this information, the original question remains. What was the pressure used? Even if it is not a quantitative answer, some comment regarding the amount of pressure must be done.
Regarding point 4, the authors answer with:
“The line in fig. 4, just joins the experimental points.”
Still no mention of this is made on the manuscript.
Regarding point 5, the authors recognize that the attempt time is material dependent, still, that information is not made available in the manuscript. Furthermore, when one states the interval 10^-12 and 10^-9, this is a typical interval and does not imply 1x10^-9. A factor of 4, such as 4x10^-9 could also be considered in such an interval.
Just to clarify where I stand, I am not stating that the conclusions of the authors are wrong. I am simply stating that the discussion should be more conservative, and that there are a lot of assumptions that the authors take as absolute truths instead of making reasonable assumptions which result in overall estimate and likely conclusion. With the current data and reasoning we cannot undoubtedly exclude a Neel relaxation, even though it I would say that, in fact, a Brownian relaxation is more likely.
Furthermore, I do not understand why the authors did not compute the theoretical curve using the parameters used in their assumptions and superimpose it upon the experimental data. This would further validate the assumptions that come from the Chantrel’s method. Or did the authors simply performed the approximation considering the initial susceptibility, saturation and H0 without verifying if the curve does in fact follow a Langevin curve? If this was the case the reader must know that this approximation was performed.
Moreover, as a matter of fact, Chantrel original paper also shows that his approach tends to underestimate the median diameter of the magnetic particles. This is due too several sources, namely the existence of a dead layer, which ultimately result in a larger median Diameter hence showing that one of the starting points of all the reasoning about the magnetic discussion cannot be used as a strict and accurate value.
Regarding point 6b, is should have been stated from the start that apparently the kerosene does not interact with the elastomer. This is not completely clear from the inspection of the pictures taken. Additionally, the authors state that the temperature is the room temperature, as if this issue is not relevant, still, the viscosity has an exponential dependency with the temperature, hence depending on the “room temperature” of the experiments that can affect the viscosity considerably. Particularly, in the case of kerosene, in the interval of 15-35ºC we can have a factor as high as 3x in play. Which temperature should the reader consider to be the room temperature? The room temperature in Brazil is not the same room temperature in Greenland unless the room benefits from a temperature control system. In the latter case the temperature would also be easy state.
Regarding point 6c, the authors use the attempt time to calculate the values which allegedly rule out the Neel relaxation, hence this answer results in a circular reasoning (where the conclusion used to avoid commenting the value of t0 is supported by the calculation which used the t0). Once again, I think the authors would benefit from some contention in the statements performed by their interpretation which uses a single characterization technique. For example, if the authors had TEM characterization to directly measure the diameter distribution, then less assumptions would be made, and the Neel relation could be ruled out in a more supported reasoning.
Author Response
"Please see the attachment."
